# Bioavailable Soil and Rock Strontium Isotope Data from Israel

Ian Moffat[1,2], Rachel Rudd[1], Malte Willmes[3], Graham Mortimer[2], Les Kinsley[2], Linda McMorrow[2], Richard Armstrong[2], Maxime Aubert[4,5,2] and Rainer Grün[5,2]

[1]Archaeology, College of Humanities, Arts and Social Sciences, Flinders University, Bedford Park, 5042, Australia
[2]Research School of Earth Sciences, The Australian National University, Canberra, 2600, Australia
[3]University of California Santa Cruz, California, USA
[4]Griffith Centre for Social and Cultural Research, Griffith University, Southport, 4222, Australia
[5]Australian Research Centre for Human Evolution, Griffith University, Nathan, 4111, Australia

*Correspondence to*: Ian Moffat (ian.moffat@flinders.edu.au)

**Abstract.** Strontium isotope ratios ($^{87}Sr/^{86}Sr$) of biogenic material such as bones and teeth reflect the local sources of strontium ingested as food and drink during their formation. This has led to the use of strontium isotope ratios as a geochemical tracer in a wide range of fields including archaeology, ecology, food studies and forensic sciences. In order to utilise strontium as a geochemical tracer, baseline data of bioavailable $^{87}Sr/^{86}Sr$ in the region of interest is required, and a growing number of studies have developed reference maps for this purpose in various geographic regions, and over varying scales. This study presents a new data set of bioavailable strontium isotope ratios from rock and soil samples across Israel, , as well as from sediment layers from seven key archaeological sites. This data set may be viewed and accessed both in an Open Science Framework repository (doi:10.17605/OSF.IO/XKJ5Y (Moffat et al., 2020)) or via the IRHUM (Isotopic Reconstruction of Human Migration) database.

## 1 Introduction

Strontium (Sr) isotope geochemistry has applications in many fields of research including archaeology (Bentley, 2006; Slovak and Paytan, 2012), ecology (Barnett-Johnson et al., 2008; Hobson et al., 2010), food traceability (Voerkelius et al., 2010) and forensic sciences (Beard and Johnson, 2000). Strontium is widely distributed within geological and biological materials, and the strontium isotope ratios ($^{87}Sr /^{86}Sr$) of these materials reflect the sources of strontium in the environment during their formation (Dasch, 1969). The use of traditional isotope systems such as hydrogen and oxygen systems in provenance studies is limited by their broad gradients across the Earth's surface, whereas $^{87}Sr/^{86}Sr$ ratios show predictable geographic variability determined by lithology, with some limited temporal variability (Aggarwal et al., 2008; Willmes et al., 2018). To utilise the potential of strontium isotopes as a geochemical tracer, prior knowledge of baseline data in the region of interest is required. Bedrock strontium isotope ratios are a product of the age, mineralogy, and origin of source material (Capo et al., 1998; Faure and Mensing, 2005). The amount of $^{87}Sr$ within a geological unit typically increases over time through the radioactive decay of $^{87}Rb$ to $^{87}Sr$, while the amounts of $^{84}Sr$, $^{86}Sr$ and $^{88}Sr$ remain constant (Capo et al., 1998). In

general, older rocks such as granite in continental crust have higher $^{87}$Sr /$^{86}$Sr ratios than younger rocks with a lower Rb/Sr ratio, such as oceanic basalt (Capo et al., 1998). The $^{87}$Sr /$^{86}$Sr ratios of carbonates reflect the water from which they precipitated, either seawater for marine carbonates or a local water body for non-marine carbonates (Faure, 1986; Neat et al.,

1979). Clastic sedimentary rocks typically have $^{87}$Sr /$^{86}$Sr ratios which reflect the lithology of their source material, tempered by the relative ease with which minerals containing strontium are removed by weathering (McDermott and Hawkesworth, 1990). The $^{87}$Sr /$^{86}$Sr ratios observed for geological units of varying compositions and ages are discussed further by Capo et al. (1998), Faure and Mensing (2005), Faure and Powell (1972), and others.

The strontium isotope composition of regolith is principally derived from local weathering of bedrock (Capo et al., 1998),

but may be augmented by windblown marine, mineral and anthropogenic aerosols, different mineral weathering rates, transportation of regolith and pore water chemistry (Frumkin and Stein, 2004; Goede et al., 1998). Bioavailable strontium is the component of strontium present in an environment which is available for incorporation into biological systems. Regolith, in combination with water, is the principal source of bioavailable strontium, although processes including precipitation, sea spray and fertiliser use may also affect the isotopic composition of bioavailable strontium (Bentley, 2006; Frei and Frei,

2011; Price et al., 2002; Slovak and Paytan, 2012).

## 1.1 Strontium isotope mapping

There are range of approaches to mapping bioavailable strontium, as summarised below and in greater detail in Bentley (2006), Maurer et al. (2012), Slovak and Paytan (2012) and Bataille et al. (2020). Archaeological human or faunal samples are considered to be one of the best indicators of the local bioavailable range, although their geographic origin may be

uncertain (Maurer et al., 2012; Price et al., 2002). Modern faunal samples from known localities may also be used, although non-local food sources and fertiliser use need to be considered (Bentley, 2006; Maurer et al., 2012). Soils, plants, and water may also be measured for local bioavailable strontium isotope ratios (Evans et al., 2010; Maurer et al., 2012; Price et al., 2002). While soils and plants average strontium over a smaller area, ground water and surface waters may provide an estimate of bioavailable strontium over a wider area, depending on catchment size (Evans et al., 2010; Willmes et al., 2014).

As discussed previously, the strontium isotope ratios of plants, soils and water may be affected by numerous factors including precipitation, sea spray and fertiliser use, which may vary over seasonal and annual timescales (Bentley, 2006; Hoogewerff et al., 2019; Price et al., 2002). Local bioavailable strontium isotope ratios may also be modelled from bedrock, considering age, mineralogy and weathering rates (Bataille et al., 2012; Bataille and Bowen, 2012). In the case of soil and rock samples, an ammonium acetate or ammonium chloride digestion method, which removes cations from electrostatically

bound pore water in addition to exchange sites on minerals and organic matter, can be used as a means of extracting the bioavailable component of these samples (Stewart et al., 1998). This approach has been found to extract between 0.1 and 62% of whole soil strontium (Chadwick et al. 2009).

At a country scale, strontium isoscapes have been developed for Mesoamerica (Hodell et al., 2004), the United Kingdom (Evans et al., 2010), France (Willmes et al., 2014, 2018) and Denmark (Frei and Frei, 2011) from a combination of bedrock,

soil, plant and water samples. Isoscapes of bioavailable strontium have also been modelled for the contiguous USA (Bataille and Bowen, 2012), Western Europe (Bataille et al., 2018; Hoogewerff et al., 2019) and the Caribbean region (Bataille et al., 2012). At a larger scale, Voerkelius et al. (2010) sampled mineral waters across Western Europe to predict bedrock $^{87}Sr/^{86}Sr$ but found local processes to result in unexpected values.

## 1.2 Strontium isotope studies and mapping in Israel

Previous studies in Israel and across the Levant region have utilised strontium isotope analyses in a variety of applications in addition to undertaking small scale mapping of bioavailable strontium. Herut et al. (1993) measured the $^{87}Sr/^{86}Sr$ ratios of rainwater samples across Israel and used these results in conjunction with the chemical composition of the sample to determine the sources of Sr and soluble salts. Shewan (2004) developed a map of bioavailable strontium throughout Israel using modern faunal bones and grass samples, for comparison to bones from archaeological sites in the region. The work of 75 Shewan (2004) and Perry et al. (2008), who measured $^{87}Sr/^{86}Sr$ from archaeological faunal dental samples in western Jordan, was collated to create a bioavailable strontium map for Israel and Jordan, combining modern flora and fauna with material from archaeological sites (Perry et al., 2009). These studies highlight several distinct provinces in the region, with samples along the Jordan Valley distinguished from those in the Eastern highlands of Israel and the Western Highlands in Jordan, as well as several smaller areas with varying isotopic signatures in the north of the study area (Perry et al., 2009). Rosenthal et 80 al. (1989) report the strontium isotope composition of gastropod shells to reconstruct water sources in the Dead Sea Rift area of the Jordan Valley. Spiro et al. (2011) measured $^{87}Sr/^{86}Sr$ ratios of water and mollusc shells in the Hula catchment in the Upper Jordan Valley, adjacent to the Golan Heights, and found distinct water sources and aquifers. Stein et al. (1997) sampled water and sediments around the Dead Sea to determine water sources and the evolution of the Dead Sea and its precursor, Lake Lisan, identifying two distinct periods of lake evolution. Hartman and Richards (2014) sampled plants and 85 invertebrates in Northern Israel and the Golan Heights to produce a map of modern bioavailable strontium isotope ratios, and to investigate potential sources of variability, including inter-site variability and the influence of precipitation. Arnold et al. (2016) sampled plants in the vicinity of Tell es-Safi/Gath to create a local bioavailable strontium map as a baseline for use in interpreting the mobility of domestic animals from archaeological sites in the region. Moffat et al. (2012) undertook spatially resolved strontium isotope analysis using laser ablation techniques to demonstrate annual migration using a bovid tooth from 90 the archaeological site of Skhul. These studies have highlighted the value and potential of bioavailable strontium mapping in this region, and the data presented in this study aims to add to this growing knowledge base.

## 2 Methods

### 2.1 Sample collection

Soil and rock samples were collected throughout Israel in September and October 2008. Sample locations were chosen 95 opportunistically, with reference to a digital 1:200000 geological map of Israel using the Old Israel Grid co-ordinate system

(Sneh et al., 1998), to provide the greatest representation of the stratigraphic units present in Israel that were accessible via road. A single rock and soil sample was taken at each site, with no replicates. Soil samples were collected from the surface, at a single point, with no attempt made to samples multiple soil horizons where present. Rock samples were taken for the principal geological unit present at each site and a brief description of the lithology made in the field, which can be viewed in the data file. Sediments from stratigraphic layers in seven archaeological sites, Amud, the Atlit Railway Bridge site, Neve David, Qafzeh, Sefunim Cave, Skhul and Tabun, were also sampled for analysis via fieldwork or from archival collections. Where possible, sediments from archaeological sites were collected for multiple stratigraphic units. Sample locations and archaeological sites are illustrated in Figure 1.

## 2.2 Analytical methods

### 2.2.1 Sample treatment

Rock and soil samples were heated to 60ºC for a minimum of 24 hours prior to sample preparation to comply with Australian quarantine procedures. After heating, rock samples were crushed to a powder using a hand piston. Rock and soil samples were passed through a 2 mm sieve, and 1 g aliquot of the sieved sample was leached by adding 2.5 ml of 1M ammonium nitrate ($NH_4NO_3$), following DIN (German Industrial Standard) ISO (International Organization for Standardization) 197310 (2009). Samples were shaken for 24 h and subsequently centrifuged at 3000 rpm for 5–10 min. 1 ml of supernatant was extracted and evaporated until dry, before being dissolved in 2 ml of 2 M high purity nitric acid ($HNO_3$).

The concentration of strontium was determined by ICP-AES (inductively coupled plasma atomic emission spectrometry). Ion exchange chromatography was then used to isolate strontium from other elements, particularly $^{87}$Rb to prevent isobaric interference with $^{87}$Sr (Dickin, 2000), using two columns filled with Eichrom strontium specific resin (prefilter resin and strontium specific resin). The strontium concentration determined by ICP-AES was used to determine the amount of sample added to the ion exchange column. Samples were diluted prior to MC-ICP-MS (multi-collector inductively coupled plasma mass spectrometer) analysis to allow for reanalysis if necessary.

### 2.2.2 Neptune MC-ICP-MS analysis

A Neptune MC-ICP-MS was used to measure strontium isotope ratios in the Environmental Geochemistry and Geochronology Laboratory at the Research School of Earth Sciences, Australian National University (ANU). The isotopes measured and Faraday cup configuration used for the analysis are shown in Table 1. Data reduction was performed offline and includes a blank, $^{87}$Rb isobar and an exponential mass bias correction.

To ensure precision, accuracy and reproducibility in the data produced, the samples were run in a sequence, which included blank and standard samples. The strontium carbonate reference material SRM987 (National Institute of Standards and Technology) was measured on the Neptune MC-ICP-MS during the sample sequence to quantify instrument drift. During the

analysis period, measurements of SRM987 ranged from $0.71012 \pm 0.00001$ to $0.71028 \pm 0.00001$, with a mean of $0.71022 \pm 0.00003$ (n=32). The soil and rock results were not calibrated based on these SRM987 measurements.

## 3 Results

60 soil samples and 48 rock samples were analysed for $^{87}Sr/^{86}Sr$. The $^{87}Sr/^{86}Sr$ ratios of the soil samples range from $0.70577 \pm 0.00001$ to $0.71020 \pm 0.00003$. The $^{87}Sr/^{86}Sr$ ratios of the rock samples range from $0.70529 \pm 0.00001$ to $0.74072 \pm 0.00001$. These results are illustrated on satellite images of Israel in Figure 2 (rock samples) and Figure 3 (soil samples). Gross lithologies, defined by observations in the field and geological maps of the region (Sneh et al., 1998), are used to group the results of Sr isotope analysis. The soil and rock samples analysed show some differences in $^{87}Sr/^{86}Sr$ ratios

between the different lithologies sampled, as illustrated in Figure 4, although there is some overlap between most lithologies. Figure 4a displays all samples analysed, while 4b displays an inset removing a high $^{87}Sr/^{86}Sr$ granite sample (site IS044) and a high $^{87}Sr/^{86}Sr$ rhyolite sample (site IS047), to display the rest of the data set more clearly. Basalt samples are generally less radiogenic in $^{87}Sr/^{86}Sr$ ratio than the other lithologies sampled for both soil and rock samples. Soil samples from carbonate (limestone, dolostone, chalk and marl), granite, kurkar (aeolian quartz sandstone with carbonate cement), rhyolite,

siliciclastic lithologies and areas with no bedrock have comparable $^{87}Sr/^{86}Sr$ ratios (Figure 4b). The median $^{87}Sr/^{86}Sr$ ratios from rock samples are slightly lower than those from soils for basalt, carbonate and kurkar samples (Figure 4b). The $^{87}Sr/^{86}Sr$ ratios measured from granite rock samples (two samples), and the rhyolite rock sample, are substantially elevated compared to other units, and compared to the soil samples from the same lithologies.

    At 43 of the sample locations, both soil and rock samples were collected, and the $^{87}Sr/^{86}Sr$ ratios of these samples are

compared in Figure 5. As with Figure 4, Figure 5a illustrates the entire data set, while Figure 5b has a granite rock sample (site IS044) and a rhyolite sample (site IS047) removed to better display the variation in the rest of the data set. The variation between soil and rock $^{87}Sr/^{86}Sr$ ratios for samples collected from the same site ranges from 0.00001 to 0.03130. This offset between rock and soil samples is also visualised in Figure 6, in which the reference line indicates where rock and soil samples have the same $^{87}Sr/^{86}Sr$ ratios for both rock and soil samples (y=x). Points which lie above this reference line have

higher $^{87}Sr/^{86}Sr$ ratios for soil samples than for rock samples, which in this study is the case for most basalt, carbonate, siliciclastic and kurkar samples. Points which lie below the y=x reference line have higher $^{87}Sr/^{86}Sr$ ratios for rock samples compared to soil samples, in this study this is the case for granite and rhyolite samples. The mean offsets from rock to soil $^{87}Sr/^{86}Sr$ ratios of samples collected from the same sites are shown in Table 2 grouped by lithology.

    Sediments from seven archaeological sites were also analysed for $^{87}Sr/^{86}Sr$; four from Amud; one from the Atlit Railway

Bridge site; two from Neve David; five from Qafzeh; three from Sefunim Cave, three from Skhul; and 16 from Tabun. These results are illustrated in Figure 7, with samples R_1024 and R_1026 from Amud highly elevated compared to the rest of the dataset (Figure 7a). Figure 7b displays a subset of the data excluding these elevated samples to better display the variability

in the rest of the samples. For the sites other than Amud, the samples from Qafzeh show the largest range in $^{87}Sr/^{86}Sr$ ratios between layers, while for Neve David, Sefunim Cave, Skhul and Tabun, most samples have similar $^{87}Sr/^{86}Sr$ ratios.

**4 Discussion**

**4.1 Comparison between rock and soil $^{87}Sr/^{86}Sr$ ratios**

The difference between the $^{87}Sr/^{86}Sr$ ratios of soil and rock sampled from the same site is in most cases greater than the analytical uncertainty (Figure 5), which may be due to inputs to the soil from sources other than bedrock, such as sea spray, irrigation or Saharan aeolian dust, or alternately due to differences in the weathering rates of minerals. Rainwater collected
across Israel has been found to contain strontium associated with sea spray, marine minerals and dust from carbonate minerals, the relative proportions of which affect the $^{87}Sr/^{86}Sr$ ratio (Herut et al., 1993).

The input of aeolian dust is expected to be particularly important to the $^{87}Sr/^{86}Sr$ ratios of these soil samples, as it has been reported that aeolian material may make up to 50% of soils formed on hard limestone rocks in Israel (Yaalon, 1997). The $^{87}Sr/^{86}Sr$ isotope ratio of dust in the region is known to have varied over time, which is an important consideration in the use
of modern isoscapes for the interpretation of archaeological studies. $^{87}Sr/^{86}Sr$ ratios of dust in this region varied from 0.711–0.712 during MIS 2 and 4, and 0.709–0.710 during MIS 1, 3 and 5 (Haliva-Cohen et al., 2012). The delivery of aeolian dust is also affected by climate variations (Frumkin et al., 2011), with Sr isotope ratios found to increase at major climate transitions that correspond to sapropel formation in the Mediterranean (Stein et al., 2007). Speleothem analyses have shown that colder and drier conditions are associated with higher Sr isotope ratios due to an increase in sea spray and aeolian dust
(Ayalon et al., 1999; Frumkin and Stein, 2004).

**4.2 Sediment $^{87}Sr/^{86}Sr$ ratios from archaeological sites**

The sediments analysed from archaeological sites in this study are mostly in carbonate units (limestone, dolostone, marl, chalk, chert) which are Mid-Eocene to Albian-Cenomanian in age, while the Atlit Railway Bridge site is located in natural caves formed in Quaternary kurkar (aeolian quartz sandstone with carbonate cement) on the coastal plain (Porat et al., 2018).
The single sample analysed from the Atlit Railway Bridge site has a $^{87}Sr/^{86}Sr$ ratio of $0.70952 \pm 0.00001$, which is slightly elevated compared to the other kurkar soils analysed in this study, which range from $0.70894 \pm 0.00001$ to $0.70934 \pm 0.00013$ (n=4). Soils associated from carbonate units in this study have $^{87}Sr/^{86}Sr$ ratios which range from $0.70851 \pm 0.00011$ to $0.70925 \pm 0.00001$ (n=35), although not all of the sediments from archaeological sites in this study from carbonate units fall within this range. Sediment samples from Tabun and Qafzeh in particular have variable $^{87}Sr/^{86}Sr$ ratios. Samples from
stratigraphic unit B1 at Amud are the most varied; samples R_1024 and R_1026 have highly elevated $^{87}Sr/^{86}Sr$ ratios compared to samples R_1025 and R_1027. Unit B at Amud is composed of interbedded calcareous silt, calcareous concretions and ash derived from anthropogenic activity (Madella et al., 2002). These elevated $^{87}Sr/^{86}Sr$ ratios are likely due to the ash component in these sediments, and the variable nature of the interbedded sediments at this site.

## 4.3 Comparison with regional datasets

Several previous studies analysing strontium isotopes have been undertaken in Israel, particularly in Northern Israel and the Golan Heights region. The results obtained in this study of soils derived from basalts in the Golan Heights region have $^{87}Sr/^{86}Sr$ ratios in the range of 0.70577 to 0.70681 for samples (sample numbers: 175, 203, 204, 205 and 216) with robust geological provenance. Basalt rock samples (sample numbers: 119, 143, 144, 147 and 278) in this study from the Golan Heights region have $^{87}Sr/^{86}Sr$ ratios ranging from 0.70529 to 0.70681. For the same region, Shewan (2004) report $^{87}Sr/^{86}Sr$

ratios ranging from 0.70529 to 0.70571 (n=4), while the results of Spiro et al. (2011) of $^{87}Sr/^{86}Sr$ measured from water samples in the Hula Valley and Golan Heights region range from 0.70467 to 0.70790 (n=37). The small difference between the results of this study and those of Shewan (2004) may reflect the limited number of samples analysed, as the larger data set of Spiro et al. (2011) encompasses the results of both this study and Shewan (2004). Weinstein (2006) measured bedrock $^{87}Sr/^{86}Sr$ from basalt units in the region, ranging from 0.7031 to 0.7034, lower than those measured in this study. Hartman

and Richards (2014) measured $^{87}Sr/^{86}Sr$ ratios from plants and invertebrates, and for samples taken from areas with basalt bedrock, ligneous (woody) plants were found to have $^{87}Sr/^{86}Sr$ ratios ranging from 0.70456 to 0.70851, non-ligneous plants ranging from 0.70473 to 0.70872, and invertebrates from 0.70494 to 0.70868. Rosenthal et al. (1989) summarise $^{87}Sr/^{86}Sr$ isotope sampling of basalt derived groundwater in the Jordan Valley, with $^{87}Sr/^{86}Sr$ ratios of 0.7045 to 0.705. Rainwater in the Golan Heights region was found to have a $^{87}Sr/^{86}Sr$ isotope composition in the range of 0.70804 to 0.70923 (Herut et al.,

1993). The wide range of $^{87}Sr/^{86}Sr$ isotope results in the Golan Heights region may be explained by the rapid depletion of Sr from rocks and soil in this region due to weathering and the large contribution of aeolian dust to soil profiles (Singer, 2007: 202-206).

Carbonate units were the most sampled lithology in this study, due to their widespread distribution across the country. This lithology groups a range of carbonate units (limestones, dolostones, chalk and marl), and the $^{87}Sr/^{86}Sr$ ratios of rock samples

range from 0.70733 to 0.70911, while soil $^{87}Sr/^{86}Sr$ ratios range from 0.70803 to 0.71020. Hartman and Richards (2014) report bedrock $^{87}Sr/^{86}Sr$ ratios from Northern Israel for carbonate units to range from 0.7073 to 0.7078, comparable to the lower range of rock samples measured in this study. Ligneous plant samples have $^{87}Sr/^{86}Sr$ ratios ranging from 0.70789 to 0.70910, non-ligneous plants from 0.70791 to 0.70919 and invertebrates from 0.70807 to 0.70876 (Hartman and Richards, 2014), comparable to the results of this study from rock and soil samples. Rosenthal et al. (1989) summarise $^{87}Sr/^{86}Sr$

isotopes from carbonate derived water sources in the Jordan Valley as having $^{87}Sr/^{86}Sr$ ratios ranging from 0.7070 to 0.7080. Arnold et al. (2016) sampled plants in the vicinity of Tell es-Safi/Gath to create a local bioavailable strontium map as a baseline for use in interpreting the mobility of domestic animals from archaeological sites in the region. Of the 10 plant samples collected, four were from kurkar soils, with $^{87}Sr/^{86}Sr$ ratios ranging from 0.70863 to 0.70881. The kurkar soil $^{87}Sr/^{86}Sr$ ratios reported in this study are slightly more elevated, ranging from 0.70894 to 0.70934. Hartman and Richards

(2014) also measured $^{87}Sr/^{86}Sr$ ratios from plants and invertebrates from kurkar units, ligneous plants ranged from 0.70899 to 0.70903, non-ligneous plants ranged from 0.70900 to 0.70903, and two mollusc shells were measured with $^{87}Sr/^{86}Sr$ ratios

of 0.70897 and 0.70904, comparable to the kurkar soil samples reported in this study. Coastal plain grasses were analysed by Shewan (2004) in the region of El Wad cave, south of Haifa. These $^{87}Sr/^{86}Sr$ ratios ranged from 0.70886 to 0.70965, with an outlying $^{87}Sr/^{86}Sr$ ratio of 0.71003 attributed to influence from the road base (Shewan, 2004). The results from the kurkar samples in this study are comparable to those of coastal plain grasses as reported by Shewan (2004).

Samples from granite, rhyolite and siliciclastic lithologies, as well as samples from regions with no bedrock, are also reported in this study. Hartman and Richards measured $^{87}Sr/^{86}Sr$ ratios from plants growing in alluvium, with a range of 0.70712 to 0.70830 for combined ligneous and non-ligneous (n=4) plants. From this study, samples sites where no bedrock was found had $^{87}Sr/^{86}Sr$ ratios ranging from 0.70783 to 0.70952, comparable to the results of Hartman and Richards (2014). There are no comparable studies which sample bioavailable $^{87}Sr/^{86}Sr$ for the granite and rhyolite samples included in this study.

## 5 Conclusion

This data set represents a substantial contribution of bioavailable soil and rock $^{87}Sr/^{86}Sr$ ratios for Israel, to complement and build on previous research. The $^{87}Sr/^{86}Sr$ results are shown to be principally controlled by lithology and are in broad agreement with previous, smaller scale, studies in the region. Soil and rock $^{87}Sr/^{86}Sr$ ratios from the same site are generally offset, and determining the mechanisms behind this offset could be the focus of future research to supplement this study.

## 6 Data availability

The data set can be viewed and downloaded on the IRHUM (Isotopic Reconstruction of Human Migration) database (http://www.irhumdatabase.com). The IRHUM database architecture and functionality are described by Willmes et al. (2014). The data is also available at the Open Science Framework data repository "Data Associated with Bioavailable Soil and Rock Strontium isotope data from Israel" https://doi.org/10.17605/OSF.IO/XKJ5Y.

## 7 Acknowledgements

This research was supported by Australian Research Council Discovery grants DP0664144 and DP110101417 to Professor Rainer Grün. Dr Ian Moffat is the recipient of Australian Research Council Discovery Early Career Award [DE160100703] funded by the Australian Government.

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

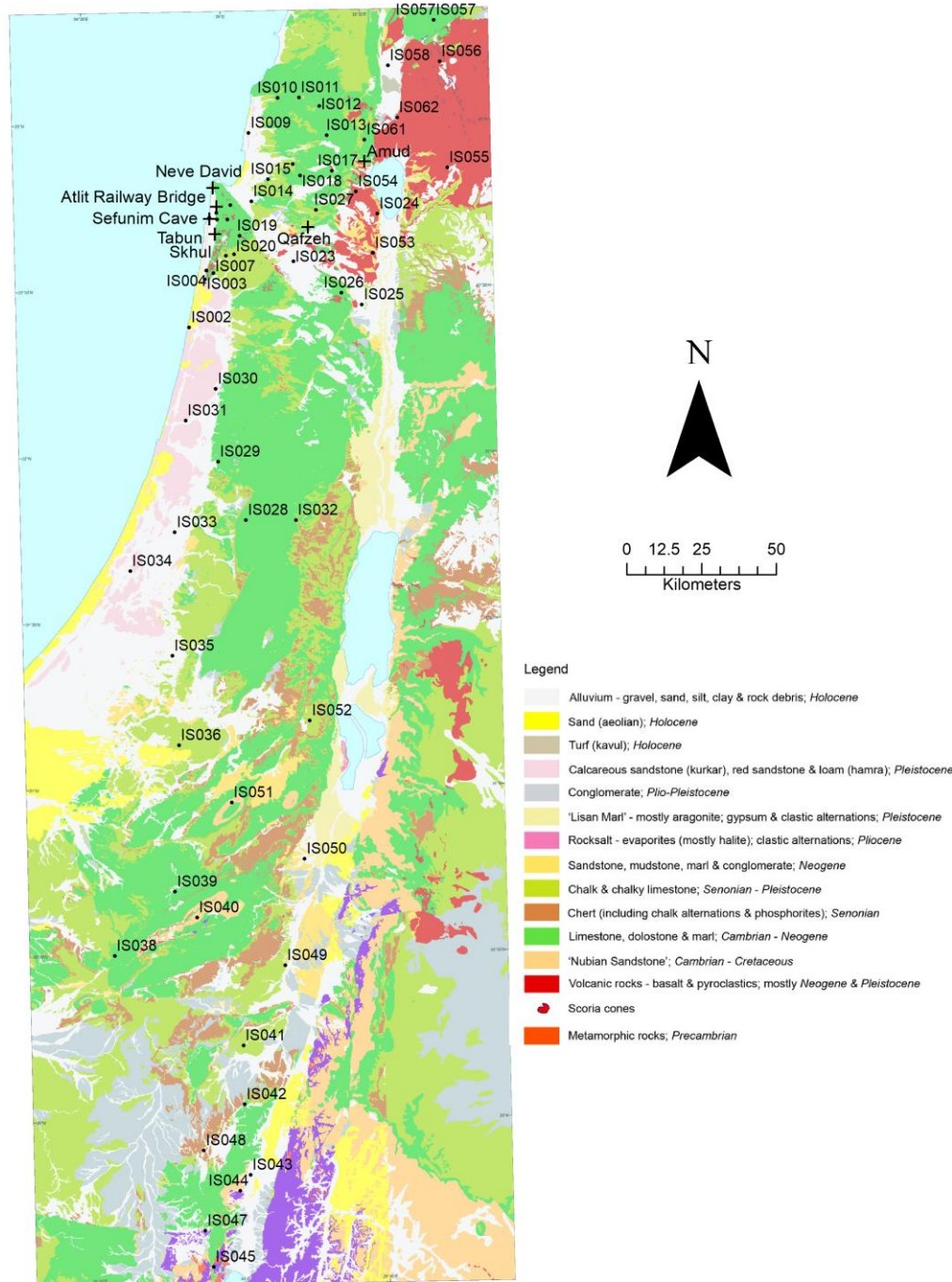

**Figure 1: Sampling locations overlain on lithological map of Israel, adapted from Sneh and Rosensaft (2014)**

**Table 1: Standard cup configuration and analysed masses (amu or isotope mass) employed for solution strontium isotope analysis on the Neptune MC-ICP-MS at RSES.**

| L4 | L3 | L2 | L1 | C | H1 | H2 | H3 | H4 |
|---|---|---|---|---|---|---|---|---|
| 82.5 | $^{83}$Kr | 83.5 | $^{84}$Sr | $^{85}$Rb | $^{86}$Sr | 86.5 | $^{87}$Sr | $^{88}$Sr |


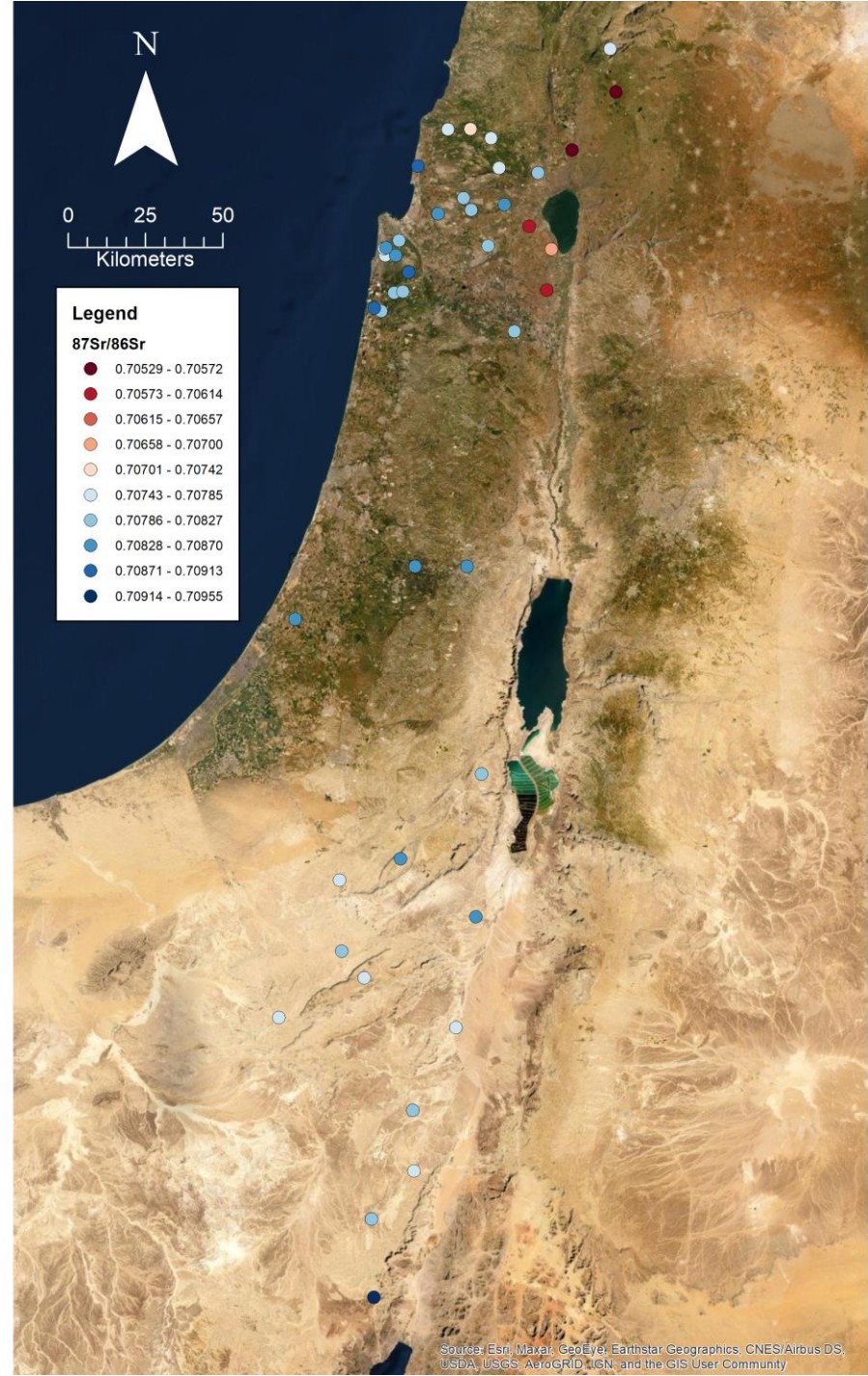

**Figure 2: Bioavailable ⁸⁷Sr/⁸⁶Sr of rock samples, excluding samples with high ⁸⁷Sr/⁸⁶Sr ratios**


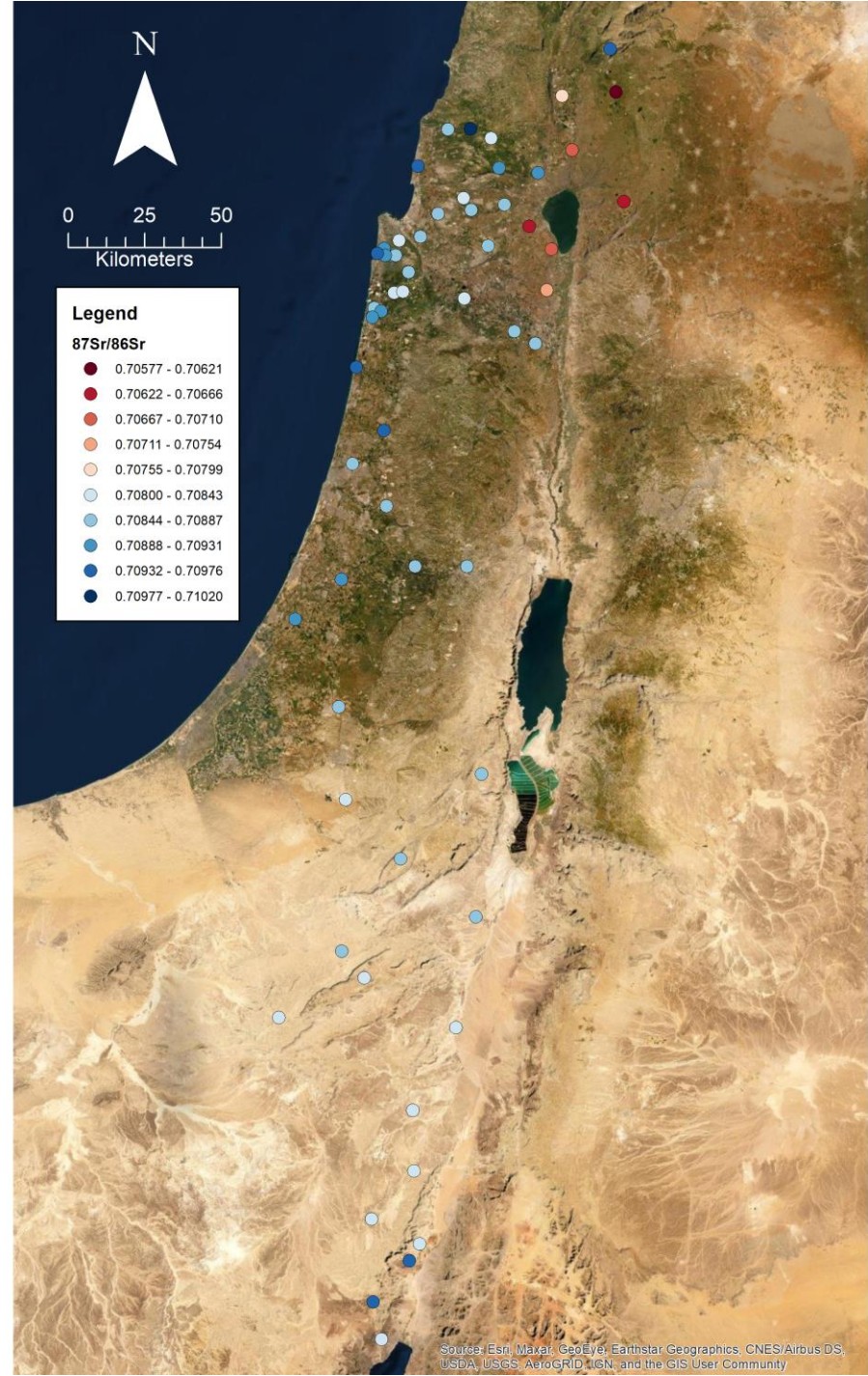

**Figure 3: Bioavailable $^{87}$Sr/$^{86}$Sr of soil samples**

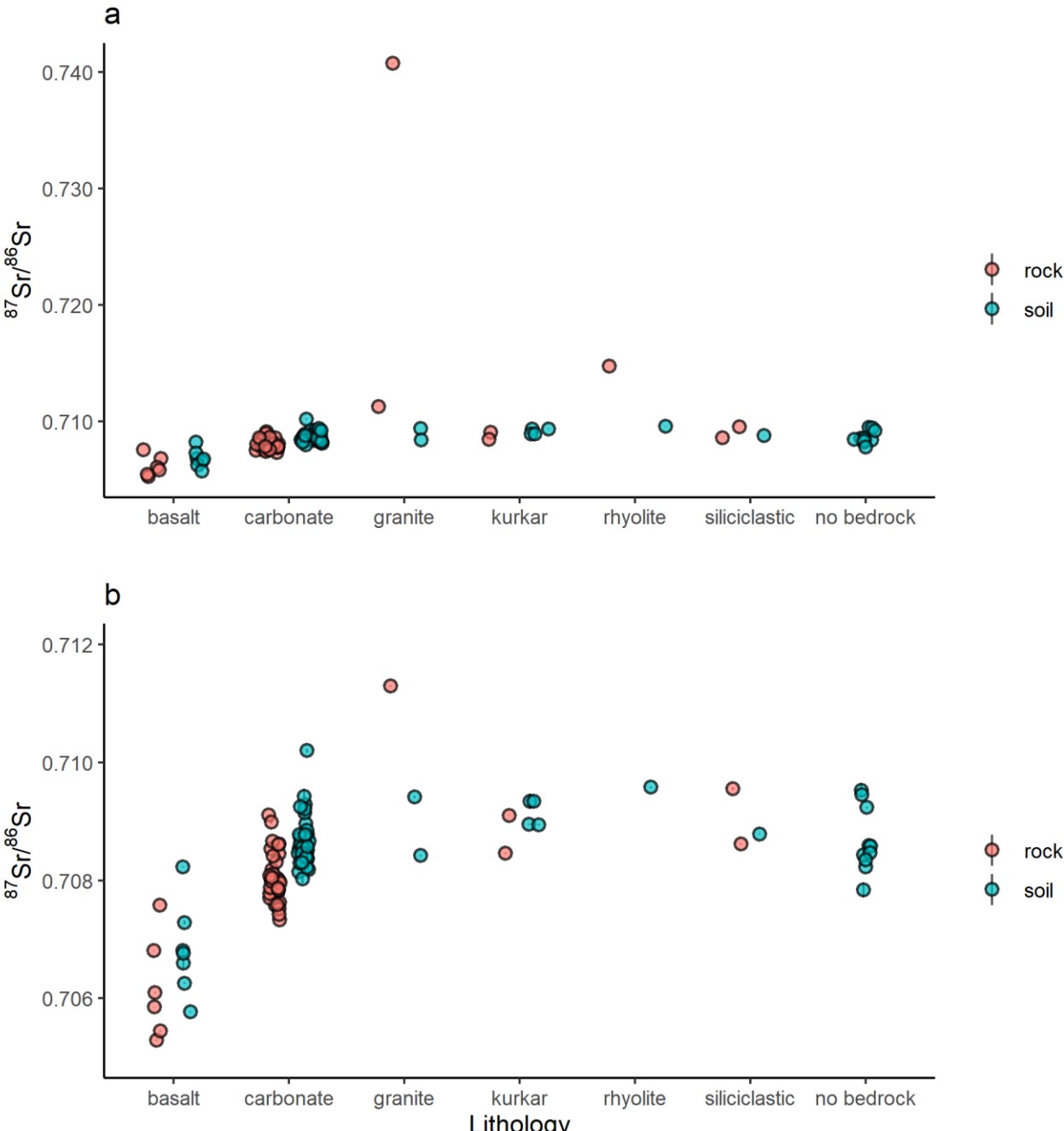


**Figure 4: Bioavailable $^{87}$Sr/$^{86}$Sr of rock and soil samples in Israel, grouped by gross lithology observed in the field. Points indicate individual samples. a) all data, b) subset of data with the two highest $^{87}$Sr/$^{86}$Sr ratio samples, one each from the granite (site IS044) and rhyolite (site IS047) lithologies, removed to better display variability between other samples.**

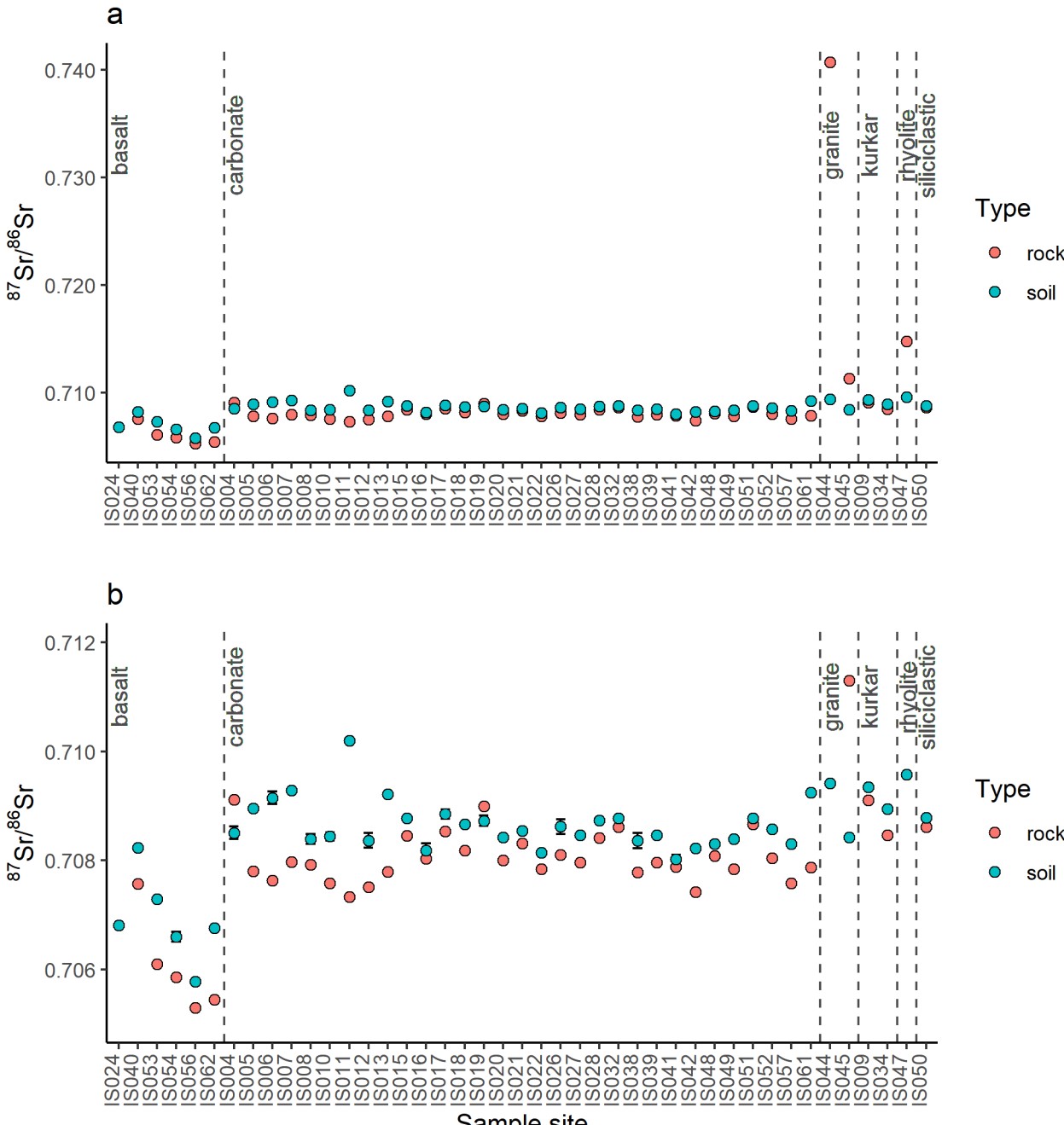

Figure 5: Bioavailable $^{87}Sr/^{86}Sr$ of rock and soil samples collected from the same sample locations displayed with $2\sigma$ error bars. Sample locations are grouped based on gross lithology observed in the field. a) all data, b) subset of data with high $^{87}Sr/^{86}Sr$ granite sample (site IS044) and rhyolite sample (site IS047) removed to display variability between other samples.

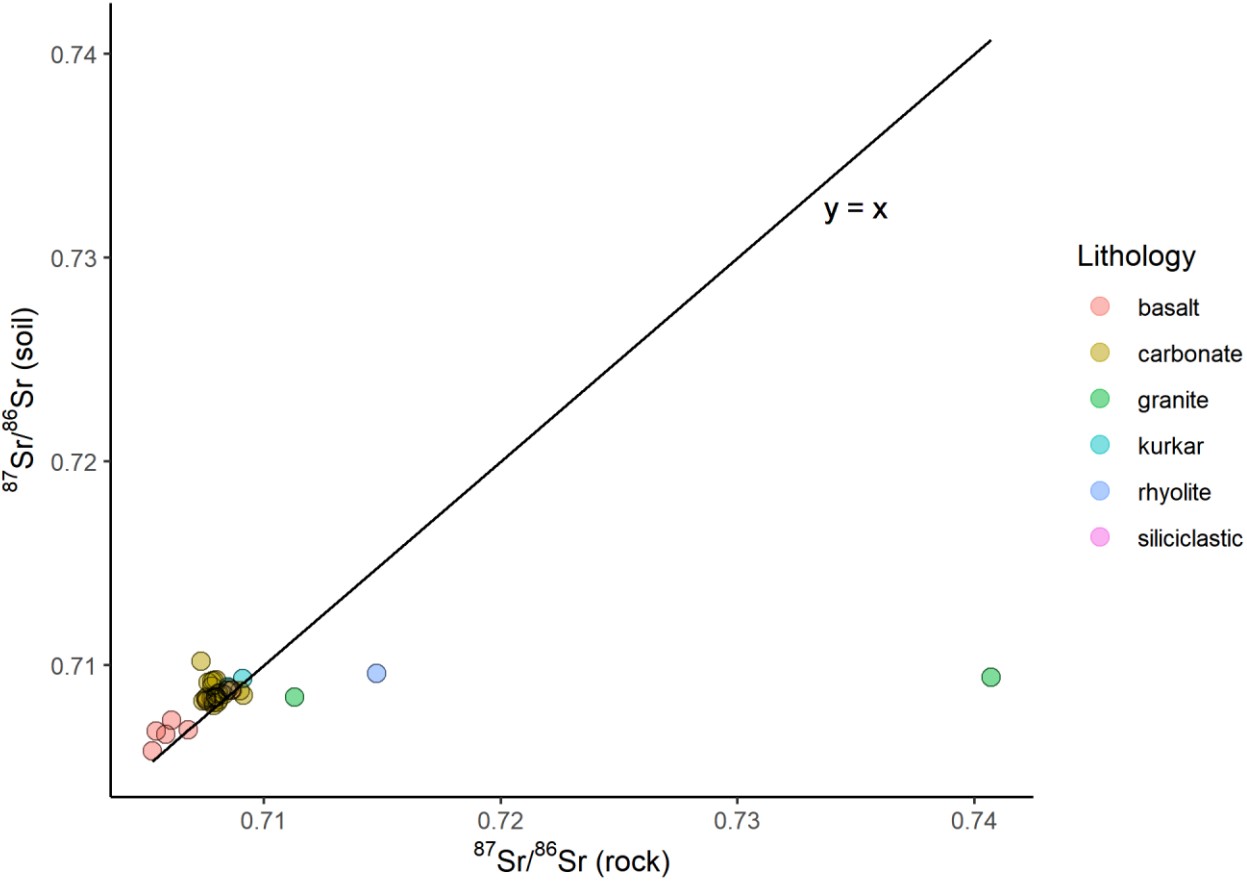

**Figure 6: Bioavailable $^{87}Sr/^{86}Sr$ of rock and $^{87}Sr/^{86}Sr$ of soil samples collected from the same sample locations. Sample locations are**
**grouped based on gross lithology observed in the field. Points which lie along the reference line (y=x) have the same $^{87}Sr/^{86}Sr$ ratios**
**for both soil and rock samples from the same sample site, while points above the line indicate a higher soil $^{87}Sr/^{86}Sr$ ratio than**
**rock, and points which lie below this line indicate a rock $^{87}Sr/^{86}Sr$ ratio higher than the soil sample from the same site.**

**Table 2: Mean offset by lithology from rock to soil $^{87}Sr/^{86}Sr$ ratios collected from the same sample locations.**

| Lithology (no. of sample sites) | Basalt (6) | Carbonate (31) | Granite (2) | Kurkar (2) | Rhyolite (1) | Siliciclastic (1) |
|---|---|---|---|---|---|---|
| Mean offset in rock vs. soil $^{87}Sr/^{86}Sr$ | 0.00073 | 0.00061 | -0.01709 | 0.00036 | -0.00517 | 0.00017 |


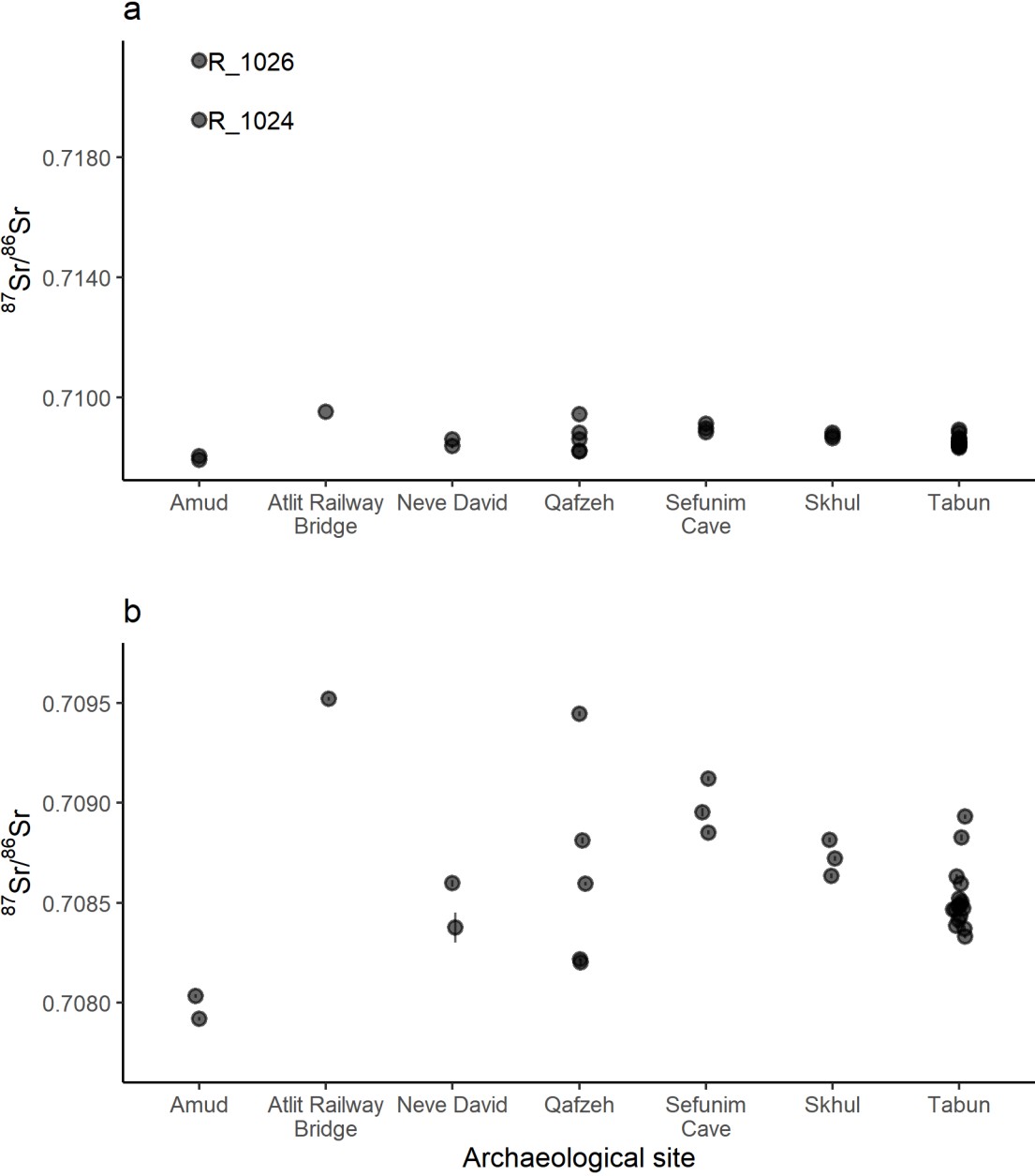

**Figure 7:** $^{87}Sr/^{86}Sr$ of soils collected from archaeological sites in Israel, a) all data, b) subset excluding the highest $^{87}Sr/^{86}Sr$ ratios of samples R_1024 and R_1026 from Amud to display the range of variability in the rest of the data set.