# Peer review of "Bioavailable Soil and Rock Strontium Isotope Data from Israel"

_Earth System Science Data, 2020_

## Referee Comment (RC1) · Jurian Hoogewerff (Referee) · 13 Jul 2020

Review of ESSD-2020-162

I thank the authors for submitting and sharing new data which contributes to the increasing world database of Sr isotope data for many applications.

The article overall is somewhat short but well written, and the data generation is well executed. However, in my opinion the discussion (and manuscript) could be strengthened in a number of ways: for example, by including the data in line 85 and 98, and/or by including the data from the literature in new combined maps and graphs.

Regarding the abstract: in the first line the surprising and rather concerning statement is made that bones and teeth are made of biogenic carbonate. Although animal and

human bone and teeth have some carbonate (for most around 4%) the majority of bone and teeth is made of bio-apatite which is a calcium phosphate mineral and Sr exchanges with Ca in either the phosphate or minor carbonate.

Regarding the sampling; it would help if the authors could clarify how soil and rock samples were taken at each site. Was it just a single soil and single rock sample at each site? Were replicate samples taken? If so, what was the variation at one soil/rock site? Was the soil sample a composite of a square meter or something else? This is important in relation to the spread of values observed in figure 4 to determine if the observed variation is very local or characteristic for a whole lithology. The choice of sampling sites is also not properly explained. Is this to fill in gaps from the literature or are the sites chosen for representative lithology or convenience? The maps would benefit from showing the locations of previous literature sampling points. What is also missing is a description of the mineralogy of at least the rock samples, and evidence that confirms that the collected rock samples match the expectation from the lithological map mentioned.

Regarding the results: when reporting scientific results, one should always consider the number of significant figures. In the text it is not clear what the quoted uncertainties entail, presumably single standard deviations of a single measurement (although figure 4 mentions 2sd) ? If so, proper reporting of the for example a value of 0.710199 $\pm$ 0.000034 should be as 0.71020 $\pm$ 0.00003. The reporting of extremely "precise" numbers for Sr isotopes in soil samples suggest very well constraint values in the field, but proper analyses of replicates mostly shows the real variation in the field to be in the 3rd decimal of Sr isotopes. This is extremely important in forensic applications as to over-estimate precision (and accuracy) might lead to wrongful conclusions.

Line 98: mentions that also elemental analysis was performed. Why has that data not been used in the discussion of the data? It might elucidate important processes like the mentioned influences of seaspray and dust? Same for data mentioned in line 85. Using this data like the pH would probably strengthen the discussion.
Line 110: It is custom to mention the value of SRM987 during the measurement period and explain if any normalization was applied?

Line 117: Figure 3 does not really show statistical "correlations" with lithology. The graph assumes a high familiarity of the reader with Sr isotope systematics, which is unfair on others, thus the text should explain why a trained isotope geochemist "sees" some confirmation of expectations related to lithology and/or geological age. Has whole rock/soil XRF analysis been performed on the samples? This would help to better define the lithology.

Line 129: significant figures?

Line 138: The text refers to "error", but what error is meant here. As alluded to above there is a major difference between instrument or method error versus variation in the or a field. It would be very helpful to know what the variation was in either soil or rock at any of the sites. Previous work, using large amount of replicates within a lithology, ( see Voerkelius et al) has shown that the variation of Sr isotope in a local lithology is much bigger than the analytical variation.

Line 140-148: Interesting mention of the variability of the dust input but how stable is the Sr isotope signal on an annual basis (food authentication of forensics) or on an archeological time scale? Would be interesting to get the authors opinion about that. In addition, it would be good to try to get a better hold on the reason behind the "offset". The authors already mention seaspray and dust, but a third component could be irrigation, which in many parts of Israel is water from Yam Kinneret and piped around the country. Noting that water from Yam Kinneret has shown stable isotope fractionation of Sr isotopes (see literature DOI: 10.1016/j.gca.2017.07.026) so it could give an extra marker for irrigation water contribution. Noting that the authors bracketed their measurements on the Neptune with ample standards they might be able to recover $88Sr/86Sr$ data from the soil measurements. Worth a try!

Line 155-185; please round the Sr isotope figures to max 4 decimals as due the limited

sample numbers the present numbers are again over-representing the accuracy

Line 194-195: The conclusion statement that the dataset is "comprehensive" is debatable as only 40 sites were sampled (on average one sample per ∼550km2), and the sampling map clearly show large gaps. But it is a good start and complements other work. In addition, it would be good to investigate more what is the reason for the "offset"

Figure 1 and 2: What is the rationale behind the cut of levels for the colors? Other authors have used "packages" or deciles. Maybe it would be beneficial to add sampling points from the other discussed literature sources? Why is a satellite image used and not the geological/lithological map of Israel, as that would relate more to the choice of sites?

Figure 3: please add "n" numbers of samples in each lithology. Is it really 2 for granite? If so the box and whisker is very tentative, probably too tentative to present. A box and whisker plot gives information about quartiles and one could argue that that at least ∼7 observations would be a minimum to make any statements a such.

Figure 4: errors bars not visible. What 2sd values were use? Instrument sd's? or method sd's.

Table 2: best to report only significant figures.

---

## Referee Comment (RC2) · Anonymous Referee #2 · 18 Aug 2020

Review of Mofat et al. manuscript titled: Bioavailable soil and rock strontium isotope data from Israel

I would like to say that the manuscript innovates by measuring bioavailable strontium isotope ratio of paired soils and underlying bedrock across Israel. The nature of the study is exploratory, and as such it can be used as archival dataset for future studies. Still the manuscript needs to go through substantial revisions before it can be approved for publication because of the following major concerns: Methods: mapping scale is questionable, consideration for sampling locations is unclear; soil and rock sampling strategy (surface, depth) is missing. Sampling permit? Sample processing and data quality assurance is only partially described. Results: deflation of variability in the region because of a couple of igneous rock samples differences in the range of 0.7050

– 0.7090 become almost invisible. The incorporation of rhyolite and quartz makes little practical sense, those are only found in very localized hyper arid region in the North West tip of the Arabian plate (AKA Eilat Mountains). Hard to accept elevated 87Sr/86Sr ratio both absolutely >0.7092 (in sedimentary bedrock soils) and relatively >0.7058 in volcanic rock are not questioned by the authors. I fear the case of spec resin contamination, check blanks results.

Detailed comments: Line 11: Abstract. "Strontium isotope ratios of biogenic carbonates such as bone and teeth". This sentence is erroneous; bone and teeth are made of carbonate apatite. The strontium is not found in carbonate, it substitutes calcium in apatite. Line 84: The use of 1:200,000 scale geological map is unclear to me, the Geological Survey of Israel provide much more precise and updated 1:50,000 scale geological maps. Lines 83– 86: were soil samples collected from the surface? Was there a consideration of soil depth or removal of topsoil? How were sampling locations determined? It is unclear if samples were collected from undisturbed environments, or perhaps from anthropogenically affected areas (agriculture, roads, industrial and residential pollution). If the sampling locations were chosen in protected areas (parks, and nature reserves) was sampling permission granted to the authors? Line 92: "rock samples were crashed to a medium powder. . .", I suggest taking out the arbitrary word: medium from the sentence. Lines 96-97: ". . . evaporated until dry, before being dissolved in 2ml of 2M high purity nitric acid, evaporated until dry and then dissolved in 2ml of 2M high purity nitric acid". I might misunderstand but It looks like the same step was repeated twice? Lines 98 – 99: Strontium concentration is measured by ICP-AES, what is the error on the measurement (I'm used to concentration measurement with ICP-MS with higher precision)? Lines 109 - 110: what is the analytical error on the measurements? Line 113: Sr isotope ratio results should be reported in up to 4 positions from the decimal point (any additional position is meaningless). 0.705772 should be 0.7058 $\pm 2\sigma$. It is customary to report strontium isotope ratio $\pm$ 2 standard deviations. Line 117: 87Sr/86Sr is a simple ratio. The use of the term "value" is meant specifically in the stable isotopes terminology to describe a normalized isotopic ratio (a ratio in a sample corrected against a ratio in a standard). Correct throughout the text. Line 138 – 140: The highly radiogenic 87Sr/86Sr ratios of Saharan dust reported from Krom et al. 1999 are measured on silicious grains, those are not bioavailable! For bioavailable strontium isotope ratios see Herut et al. 1993 Doi: 10.1016/0012-821x(93)90024-4. For atmospheric contribution see Hartman and Richards, 2014 http://dx.doi.org/10.1016/j.gca.2013.11.015; finally, for past changes in bioavailable Saharan dust contribution see high resolution data from Soreq Cave, Israel by Ayalon et al. 1999 doi: 10.1191/095968399673664163. I also suggest the authors to read again Cohen-Haliva et al. 2012 doi:10.1016/j.quascirev.2012.06.014 they specifically refer to silicate vs. carbonate strontium sources. Lines 158 – 160: Hartman and Richards 2014 did not measure bedrock 87Sr/86Sr ratios from basalt units. Line 179: Kurkar soil with 87Sr/86Sr >0.7092 (modern seawater ratio) is highly unlikely. Quoting Shewan 2004 in Lines 183 – 184 as comparable result 0.7097 is equally problematic. At least Shewan question sampling location as possible explanation for exceptionally radiogenic ratio of 0.7100. Figure 1: the volcanic bedrock 87Sr/86Sr ratios look problematic (0.7058 – 0.7063) – see Weinstein et al. 2006 10.1093/petrology/egi1003, who measured consistent bulk bedrock ratios between 0.7032 – 0.7034 across Pliocene – Pleistocene basalts in Israel. Is there a valid explanation to such a large discrepancy? When it comes to volcanic bulk and biogenic fraction, I do not think there should be a big difference. Figure 2: check all the ratios between 0.7092 – 0.7095 excluding those coming from soils that developed over the Arabian plate igneous rocks (southernmost brown symbols on the map). Those are impossible ratios. Figures 3+4: the inflation in the scales caused by the display of rhyolite (n=1) and granite (n=1) causes a complete deflation of the rest of the dataset. It is not surprising the authors treat the rest of the dataset as homogeneous.

---

## Author Comment (AC1) · 6 Oct 2020

Reviewer Comment #1: I thank the authors for submitting and sharing new data which contributes to the increasing world database of Sr isotope data for many applications. The article overall is somewhat short but well written, and the data generation is well executed. However, in my opinion the discussion (and manuscript) could be strengthened in a number of ways: for example, by including the data in line 85 and 98, and/or by including the data from the literature in new combined maps and graphs.

Author Response: The authors thank the reviewer for their constructive feedback in their review of this manuscript. These general comments are addressed later in this document as they are repeated below. The length of the paper is comparable to other

data papers, but in an effort to strengthen the manuscript and discussion, strontium isotope ratios from the sediments of seven archaeological sites in Israel are discussed and plotted in addition to the soil and rock samples included in the original submission.

Reviewer Comment #2: Regarding the abstract: in the first line the surprising and rather concerning statement is made that bones and teeth are made of biogenic carbonate. Although animal and human bone and teeth have some carbonate (for most around 4%) the majority of bone and teeth is made of bio-apatite which is a calcium phosphate mineral and Sr exchanges with Ca in either the phosphate or minor carbonate.

Author Response: This has been corrected, thank you for pointing out this error.

Reviewer Comment #3: Regarding the sampling; it would help if the authors could clarify how soil and rock samples were taken at each site. Was it just a single soil and single rock sample at each site? Were replicate samples taken? If so, what was the variation at one soil/rock site? Was the soil sample a composite of a square meter or something else? This is important in relation to the spread of values observed in figure 4 to determine if the observed variation is very local or characteristic for a whole lithology. The choice of sampling sites is also not properly explained. Is this to fill in gaps from the literature or are the sites chosen for representative lithology or convenience? The maps would benefit from showing the locations of previous literature sampling points. What is also missing is a description of the mineralogy of at least the rock samples, and evidence that confirms that the collected rock samples match the expectation from the lithological map mentioned.

Author Response: Thank you for your comment, this has been clarified in the text and answered below. A single rock and soil sample was taken at each site, with no replicates and the soil sample taken from a single point with no attempt to average the sample over a large area. Sample locations were chosen opportunistically to provide the greatest representation of the stratigraphic units present in Israel that were eas-

ily accessible via roads. Soil samples were collected from the topsoil and no attempt was made to sample multiple soil horizons when they were present, other than in archaeological sites where multiple stratigraphic units were samples where possible. We attempted to collect samples from undisturbed areas where there was an obvious spatial relationship between the soil and bedrock however cannot guarantee that these sites were free of anthropogenic contamination. No samples were collected in parks or nature reserves. The lithology of the rock samples and a brief description was recorded in the field, these are included in our data file as "Geology Observed in the Field" and "Geology Description from Field". We have not analysed the mineralogy of the rock or plant samples beyond what was described in the field. We agree that a combined map of all bioavailable strontium isotope analysis undertaken in Israel would be useful however the purpose of this paper was to share the data that we have collected. Further, the various other studies contain some inconsistencies in reporting their sample locations which make it challenging to integrate these data sets with ours.

Reviewer Comment #4: Regarding the results: when reporting scientific results, one should always consider the number of significant figures. In the text it is not clear what the quoted uncertainties entail, presumably single standard deviations of a single measurement (although figure 4 mentions 2sd)? If so, proper reporting of the for example a value of $0.710199 \pm 0.000034$ should be as $0.71020 \pm 0.00003$. The reporting of extremely "precise" numbers for Sr isotopes in soil samples suggest very well constraint values in the field, but proper analyses of replicates mostly shows the real variation in the field to be in the 3rd decimal of Sr isotopes. This is extremely important in forensic applications as to over-estimate precision (and accuracy) might lead to wrongful conclusions.

Author Response: Significant figures have been corrected throughout the text and figures. The errors reported are 2sd, from the measurement only as a single sample was collected at each site.

Reviewer Comment #5: Line 98: mentions that also elemental analysis was performed.

Why has that data not been used in the discussion of the data? It might elucidate important processes like the mentioned influences of seaspray and dust? Same for data mentioned in line 85. Using this data like the pH would probably strengthen the discussion.

Author Response: This comment refers to the soil colour and pH (line 85) and Sr concentration and elemental analysis. Strontium concentrations measured using ICP-OES and used to optimise our column chemistry and soil observations recorded in the field (grain size, sorting, colour, pH) have been added to the data files available online. The pH data for the soils is very uniform, ranging from 7.5 to 9.5 (although most samples are 8-9) for all samples, so has not been explored further in the text. Strontium was the only element measured, so we have removed mention that other elements were also analysed.

Reviewer Comment #6: Line 110: It is custom to mention the value of SRM987 during the measurement period and explain if any normalization was applied?

Author Response: During the analysis period, SRM987 ranged from 0.71012 $\pm$ 0.00001 to 0.71028 $\pm$ 0.00001, with a mean of 0.71022 $\pm$ 0.00003 (n=32). No normal-isation was applied to the data using these data. This information has been added to the text.

Reviewer Comment #7: Line 117: Figure 3 does not really show statistical "correla-tions" with lithology. The graph assumes a high familiarity of the reader with Sr isotope systematics, which is unfair on others, thus the text should explain why a trained iso-tope geochemist "sees" some confirmation of expectations related to lithology and/or geological age. Has whole rock/soil XRF analysis been performed on the samples? This would help to better define the lithology.

Author Response: Thank you for the comment, we have updated the introduction to clarify some of the expected 87Sr/86Sr ratios in different lithologies. We have also avoided use of the word 'correlation' where a statistical relationship is not being discussed. XRF was not performed as part of the analyses in this project and we have accepted the lithology presented on the relevant geological map for each sample location.

Reviewer Comment #8: Line 129: significant figures?

Author Response: Amended.

Reviewer Comment #9: Line 138: The text refers to "error", but what error is meant here. As alluded to above there is a major difference between instrument or method error versus variation in the or a field. It would be very helpful to know what the variation was in either soil or rock at any of the sites. Previous work, using large amount of replicates within a lithology, (see Voerkelius et al) has shown that the variation of Sr isotope in a local lithology is much bigger than the analytical variation.

Author Response: A single sample was collected from each location, so these errors do not refer to variation in the field, this is instrument error.

Reviewer Comment #10: Line 140-148: Interesting mention of the variability of the dust input but how stable is the Sr isotope signal on an annual basis (food authentication of forensics) or on an archeological time scale? Would be interesting to get the authors opinion about that. In addition, it would be good to try to get a better hold on the reason behind the "offset". The authors already mention seaspray and dust, but a third component could be irrigation, which in many parts of Israel is water from Yam Kinneret and piped around the country. Noting that water from Yam Kinneret has shown stable isotope fractionation of Sr isotopes (see literature DOI: 10.1016/j.gca.2017.07.026) so it could give an extra marker for irrigation water contribution. Noting that the authors bracketed their measurements on the Neptune with ample standards they might be able to recover $88Sr/86Sr$ data from the soil measurements. Worth a try!

Author Response: Further investigation of the influence of irrigation on strontium isotope ratios in Israel would be an outstanding addition to this research for future studies,

but unfortunately is beyond the scope of this project.

Reviewer Comment #11: Line 155-185; please round the Sr isotope figures to max 4 decimals as due the limited sample numbers the present numbers are again over-representing the accuracy.

Author Response: The data from this study has all been revised to display 5 significant figures in response to a previous comment. The data in lines 155 to 185 is from other studies, as referenced, which was reported to 5 significant figures.

Reviewer Comment #12: Line 194-195: The conclusion statement that the dataset is "comprehensive" is debatable as only 40 sites were sampled (on average one sample per âLij550km2), and the sampling map clearly show large gaps. But it is a good start and complements other work. In addition, it would be good to investigate more what is the reason for the "offset".

Author Response: The authors agree that this dataset represents an exploration into strontium isotope ratios across Israel, and future research could improve upon this data in several ways, including further investigation of the offset observed between soil and rock samples. This dataset, however, is far more comprehensive than other work that has previously been undertaken in the region, and so represents a significant contribution to this field. We have clarified this in the manuscript text.

Reviewer Comment #13: Figure 1 and 2: What is the rationale behind the cut of levels for the colors? Other authors have used "packages" or deciles. Maybe it would be beneficial to add sampling points from the other discussed literature sources? Why is a satellite image used and not the geological/lithological map of Israel, as that would relate more to the choice of sites?

Author Response: The colour intervals used were chosen using the Jenks natural breaks in ArcMap, which optimises the deviation within individual classes relative to those of other groups, but is perhaps not very intuitive for readers. This figure has been

amended such that each colour corresponds to an equal interval. When plotted on the geological map, it is almost impossible to distinguish the sampling locations, and the colours denoting the 87Sr/86Sr ratio are very difficult to see. The satellite image was used for clarity with this data, and to hopefully make this figure accessible for a range of readers. The lithological map was not one that we had considered but would be a much better option for overlaying points, particularly for displaying the sampling locations and not the results - we have developed another map which overlays the sample locations on the lithological map. Attempts were made to integrate the sampling locations from previous studies, but very few provide coordinates of sampling locations, and those which do use a different coordinate system in each instance. Combining the data from all these studies would be an excellent next step for this research, but is beyond the scope of this data paper.

Reviewer Comment #14: Figure 3: please add "n" numbers of samples in each lithology. Is it really 2 for granite? If so the box and whisker is very tentative, probably too tentative to present. A box and whisker plot gives information about quartiles and one could argue that that at least âĹij7 observations would be a minimum to make any statements as such.

Author Response: Thank you for the comment. The individual sample points were added to what was originally a boxplot in the hope of clarifying the number of samples in each lithology, but the plot has now been amended to only show the points as the boxplots are not appropriate for the dataset, and so hopefully this figure is now clearer.

Reviewer Comment #15: Figure 4: errors bars not visible. What 2sd values were use? Instrument sd's? or method sd's.

Author Response: The error bars are very small for most samples, and have been plotted in this figure, but are obscured by the points due to the scale on the y axis used to display as much of the data as possible. 'Part b' of this figure has been amended to remove the high rhyolite rock sample from site IS047 as well as the elevated granite
Interactive
comment

rock sample, which allows some of the error bars to be visible in this subset of the full data in Part a.

Reviewer Comment #16: Table 2: best to report only significant figures.

Author Response: Thank you, this has been amended.

---

## Author Comment (AC2) · 6 Oct 2020

Reviewer Comment #1: I would like to say that the manuscript innovates by measuring bioavailable strontium isotope ratio of paired soils and underlying bedrock across Israel. The nature of the study is exploratory, and as such it can be used as archival dataset for future studies. Still the manuscript needs to go through substantial revisions before it can be approved for publication because of the following major concerns: Methods: mapping scale is questionable, consideration for sampling locations is unclear; soil and rock sampling strategy (surface, depth) is missing. Sampling permit? Sample processing and data quality assurance is only partially described.

Results: deflation of variability in the region because of a couple of igneous rock samples differences in the range of 0.7050 – 0.7090 become almost invisible. The incorporation of rhyolite and quartz makes little practical sense, those are only found in very localized hyper arid region in the North West tip of the Arabian plate (AKA Eilat Mountains). Hard to accept elevated 87Sr/86Sr ratio both absolutely >0.7092 (in sedimentary bedrock soils) and relatively >0.7058 in volcanic rock are not questioned by the authors. I fear the case of spec resin contamination, check blanks results.

Author Response: The authors would like to thank the reviewer for their constructive and comprehensive review. As the concerns listed here are repeated below in the detailed comments, they are addressed individually through this document.

Reviewer Comment #2: Line 11: Abstract. "Strontium isotope ratios of biogenic carbonates such as bone and teeth". This sentence is erroneous; bone and teeth are made of carbonate apatite. The strontium is not found in carbonate, it substitutes calcium in apatite.

Author Response: Thank you for the comment, this has been amended in the manuscript.

Reviewer Comments #3: Line 84: The use of 1:200,000 scale geological map is unclear to me, the Geological Survey of Israel provide much more precise and updated 1:50,000 scale geological maps.

Author Response: We chose to use the 1:200,000 scale geological maps rather than the excellent and very detailed 1:50,000 scale maps as this research was focused on obtaining strontium isotope values from the geological units which outcrop most widely across Israel, rather than investigating geographically smaller units. Should a denser sampling program be undertaken we would certainly recommend the 1:50,000 scale maps for that purpose.

Reviewer Comments #4: Lines 83– 86: were soil samples collected from the surface? Was there a consideration of soil depth or removal of topsoil? How were sampling locations determined? It is unclear if samples were collected from undisturbed environments, or perhaps from anthropogenically affected areas (agriculture, roads, industrial and residential pollution). If the sampling locations were chosen in protected areas (parks, and nature reserves) was sampling permission granted to the authors?

Author Response: Thank you for pointing out the ambiguity in the text about sampling procedures, we have provided additional information to clarify this. To answer these questions: soil samples were collected from the surface, and a single sample was collected, with no attempt made to sample different soil horizons. Undisturbed locations were chosen where possible, and no samples were collected from parks or nature reserves.

Reviewer Comments #5: Line 92: "rock samples were crashed to a medium powder. . .", I suggest taking out the arbitrary word: medium from the sentence.

Author Response: Thank you, this had been amended.

Reviewer Comments #6: Lines 96-97: ". . . evaporated until dry, before being dissolved in 2ml of 2M high purity nitric acid evaporated until dry and then dissolved in 2ml of 2M high purity nitric acid". I might misunderstand but It looks like the same step was repeated twice?

Author Response: Thank you, this has been amended.

Reviewer Comments #7: Lines 98 – 99: Strontium concentration is measured by ICP-AES, what is the error on the measurement (I'm used to concentration measurement with ICP-MS with higher precision)?

Author Response: ICP-AES normally has an error of ∼1%. This is sufficient for the purpose of this research, which was optimising our column chemistry.

Reviewer Comments #8: Lines 109 - 110: what is the analytical error on the measurements?

[Figure]

Author Response: We aren't certain exactly what you refer to here however have reported the analytical error to 2 standard deviations for all strontium isotope measurements and have provided SRM987 standard measurements in the updated manuscripts.

Reviewer Comments #9: Line 113: Sr isotope ratio results should be reported in up to 4 positions from the decimal point (any additional position is meaningless). 0.705772 should be $0.7058 \pm 2\sigma$. It is customary to report strontium isotope ratio $\pm$ 2 standard deviations.

Author Response: The isotope ratios have been amended to 5 decimal places, following a comment from reviewer 1. The error is 2 standard deviations.

Reviewer Comments #10: Line 117: 87Sr/86Sr is a simple ratio. The use of the term "value" is meant specifically in the stable isotopes terminology to describe a normalized isotopic ratio (a ratio in a sample corrected against a ratio in a standard). Correct throughout the text.

Author Response: Thank you for your comment, this has been amended throughout the text.

Reviewer Comments #11: Line 138 – 140: The highly radiogenic 87Sr/86Sr ratios of Saharan dust reported from Krom et al. 1999 are measured on silicious grains, those are not bioavailable! For bioavailable strontium isotope ratios see Herut et al. 1993 Doi: 10.1016/0012-821x(93)90024-4. For atmospheric contribution see Hartman and Richards, 2014 http://dx.doi.org/10.1016/j.gca.2013.11.015; finally, for past changes in bioavailable Saharan dust contribution see high resolution data from Soreq Cave, Israel by Ayalon et al. 1999 doi: 10.1191/095968399673664163. I also suggest the authors to read again Cohen-Haliva et al. 2012 doi:10.1016/j.quascirev.2012.06.014 they specifically refer to silicate vs. carbonate strontium sources.

Author Response: Thank you for providing this additional, very useful, background

information about the regional strontium isotope values in the region.

Reviewer Comments #12: Lines 158 – 160: Hartman and Richards 2014 did not measure bedrock 87Sr/86Sr ratios from basalt units.

Author Response: In Table S3 of the online supplementary material from Hartman & Richards (2014), bedrock strontium isotopes are listed from basaltic units. These values were not referenced to another study, and so it was assumed that they were measured during this study. We apologise for this oversight, as the main text does refer to Weinstein et al. (2006), and this section has been amended in the text.

Reviewer Comments #13: Line 179: Kurkar soil with 87Sr/86Sr >0.7092 (modern seawater ratio) is highly unlikely. Quoting Shewan 2004 in Lines 183 – 184 as comparable result 0.7097 is equally problematic. At least Shewan question sampling location as possible explanation for exceptionally radiogenic ratio of 0.7100.

Figure 1: the volcanic bedrock 87Sr/86Sr ratios look problematic (0.7058 – 0.7063) – see Weinstein et al. 2006 10.1093/petrology/egi1003, who measured consistent bulk bedrock ratios between 0.7032 – 0.7034 across Pliocene – Pleistocene basalts in Israel. Is there a valid explanation to such a large discrepancy? When it comes to volcanic bulk and biogenic fraction, I do not think there should be a big difference.

Figure 2: check all the ratios between 0.7092 – 0.7095 excluding those coming from soils that developed over the Arabian plate igneous rocks (southernmost brown symbols on the map). Those are impossible ratios.

Author Response: For these three comments, the authors acknowledge the reviewer's concerns regarding these values and their detailed knowledge of the strontium isotope composition of these geological units, but stand by our measurements and analytical methods. The methods used to extract bioavailable strontium from soils have been found to extract between 0.1-62% of whole soil strontium (Chadwick et al. 2009, doi: 10.1016/j.chemgeo.2009.01.009), which may partially explain the differences in strontium isotope ratios between this study and others which have measured whole soil or rock samples. The aim of this data paper is not to resolve all ambiguities between our research and other studies but to present all the data collected. We thank the reviewer for this comment and we hope that this manuscript encourages further research into the bioavailable strontium isotope values of these units.

Reviewer Comments #14: Figures 3+4: the inflation in the scales caused by the display of rhyolite (n=1) and granite (n=1) causes a complete deflation of the rest of the dataset. It is not surprising the authors treat the rest of the dataset as homogeneous.

Author Response: It was not the authors intention to communicate that the rest of the dataset is homogenous, and the plots were deliberately displayed in two panels to try and show both the entire dataset and the subset without the high Sr isotope ratio samples. The granite sample was not included in the 'part b' plot in each instance, but the rhyolite rock sample has also been removed now from part b, we hope this makes the dataset clearer as shown below.

---

## Author Response (AR1)

**Bioavailable Soil and Rock Strontium Isotope Data from Israel**

Ian Moffat1,2, Rachel Rudd1, Malte Willmes3, Graham Mortimer2, Les Kinsley2, Linda McMorrow2, Richard Armstrong2, Maxime Aubert4,5,2 and Rainer Grün5,2

[revised manuscript text omitted]
 87Sr/86Sr. The 87Sr/86Sr ratios of the soil samples range from 0.70577270577 ± 0.00001100011 to 0.71019971020 ± 0.00003400003. The 87Sr/86Sr ratios of the rock samples range from 0.70529170529 ± 0.000006000011 to 0.74071874072 ± 0.00001400001. These results are illustrated on satellite images of 135 Israel in Figure 12 (rock samples) and Figure 23 (soil samples). Gross lithologies, defined by observations in the field and geological maps of the region (Sneh et al., 1998), are used to group the results of Sr isotope analysis. SoilThe soil and rock 87Sr/86Sr valuessamples analysed show some correlation with lithologydifferences in 87Sr/86Sr ratios between the different lithologies sampled, as illustrated in Figure 3.4, although there is some overlap between most lithologies. Figure 3.4a displays

- all samples analysed, while 3b4b displays an inset removing a high 87Sr/86Sr granite sample (site IS044,) and a high 87Sr/86Sr
   rhyolite sample number 134)(site IS047), to display the rest of the data set more clearly. Basalt unitssamples are generally less radiogenic in 87Sr/86Sr valueratio than the other lithologies sampled for both soil and rock samples. Soil samples from carbonate (limestone, dolostone, chalk and marl), granite, kurkar (aeolian quartz sandstone with carbonate cement), rhyolite, siliciclastic lithologies and areas with no bedrock have comparable 87Sr/86Sr valuesratios (Figure 3b4b). The median 87Sr/86Sr valuesratios from rock samples are slightly lower than those from soils for basalt, carbonate and kurkar samples (Figure 3b4b). The 87Sr/86Sr
- 145 valuesratios measured from granite rock samples (two samples), and the rhyolite rock sample, are substantially elevated compared to other units, and compared to the soil samples from the same lithologies.

At 43 of the sample locations, both soil and rock samples were collected, and the 87Sr/86Sr <del>values</del>ratios of these samples are compared in Figure 45. As with Figure 34, Figure 4a5a illustrates the entire data set, while Figure 4b5b has thea granite rock sample (site IS044) and a rhyolite sample (site IS047) removed to better display the variation in the rest of the data set. The

- 150 variation between soil and rock 87Sr/86Sr values rangeratios for samples collected from the same site ranges from 0.00000300001 to 0.03130403130. This offset between rock and soil samples is also visualised in Figure 56, in which the reference line indicates where rock and soil samples have the same 87Sr/86Sr valuesratios for both rock and soil samples (y=x). Points which lie above this reference line have higher 87Sr/86Sr valuesratios for soil samples than for rock samples, which in this study is the case for most basalt, carbonate, siliciclastic and kurkar samples. Points which lie below the y=x reference line
- 155 have higher 87Sr/86Sr valuesratios for rock samples compared to soil samples, in this study this is the case for granite and rhyolite samples. The mean offsets from rock to soil 87Sr/86Sr valuesratios of samples collected from the same sites are shown in Table 2 grouped by lithology.

Sediments from seven archaeological sites were also analysed for 87Sr/86Sr; four from Amud; one from the Atlit Railway Bridge site; two from Neve David; five from Qafzeh; three from Sefunim Cave, three from Skhul; and 16 from Tabun. These results are illustrated in Figure 7, with samples R\_1024 and R\_1026 from Amud highly elevated compared to the rest of the dataset (Figure 7a). Figure 7b displays a subset of the data excluding these elevated samples to better display the variability in the rest of the samples. For the sites other than Amud, the samples from Qafzeh show the largest range in 87Sr/86Sr ratios between layers, while for Neve David, Sefunim Cave, Skhul and Tabun, most samples have similar 87Sr/86Sr ratios.

**4 Discussion**

160

**165 4.1 Comparison between rock and soil values87Sr/86Sr ratios**

The variationdifference between the 87Sr/86Sr valuesratios of soil and rock sampled from the same site is in most cases greater than errorthe analytical uncertainty (Figure 45), which may be due to inputs to the soil from sources other than bedrock, such as sea spray, irrigation or Saharan aeolian dust (87Sr/86Sr values, or alternately due to differences in the rangeweathering rates of 0.7160–0.7192 (Kromminerals. Rainwater collected across Israel has been found to contain strontium associated with sea spray, marine minerals and dust from carbonate minerals, the relative proportions of which affect the 87Sr/86Sr ratio (Herut et

170 spray, marine minerals and dust from carbonate minerals, the relative proportions of which affect the 87Sr/86Sr ratio (He al., 1999)).

The input of aeolian dust is expected to be particularly important into the 87Sr/86Sr valuesratios of these soil samples, as it has been reported that aeolian material may make up to 50% of soils formed on hard limestone rocks in Israel (Yaalon, 1997). The  $Sr^{87}Sr/^{86}Sr$  isotope valueratio of dust in the region is known to have varied over time, which is an important consideration in

- the use of modern isoscapes for the interpretation of older sample material.archaeological studies.87Sr/86Sr valuesratios of dust in this region varied from 0.711–0.712 during MIS 2 and 4, and 0.709–0.710 during MIS 1, 3 and 5 (Haliva-Cohen et al., 2012). The delivery of aeolian dust is also affected by climate variations (Frumkin et al., 2011), with Sr isotope valuesratios found to increase at major climate transitions that correspond to sapropel formation in the Mediterranean (Stein et al., 2007). Speleothem analyses have shown that glacial periods have a higher aeolian Sr isotope contribution than interglacial periods
- 180 (colder and drier conditions are associated with higher Sr isotope ratios due to an increase in sea spray and aeolian dust (Ayalon et al., 1999; Frumkin and Stein, 2004).

**4.2 Sediment 87Sr/86Sr ratios from archaeological sites**

The sediments analysed from archaeological sites in this study are mostly in carbonate units (limestone, dolostone, marl, chalk, chert) which are Mid-Eocene to Albian-Cenomanian in age, while the Atlit Railway Bridge site is located in natural caves
 formed in Quaternary kurkar (aeolian quartz sandstone with carbonate cement) on the coastal plain (Porat et al., 2018). The single sample analysed from the Atlit Railway Bridge site has a 87Sr/86Sr ratio of 0.70952 ± 0.00001, which is slightly elevated compared to the other kurkar soils analysed in this study, which range from 0.70894 ± 0.00001 to 0.70934 ± 0.00013 (n=4). Soils associated from carbonate units in this study have 87Sr/86Sr ratios which range from 0.70851 ± 0.00011 to 0.70925 ±

0.00001 (n=35), although not all of the sediments from archaeological sites in this study from carbonate units fall within this
 range. Sediment samples from Tabun and Qafzeh in particular have variable 87Sr/86Sr ratios. Samples from stratigraphic unit
 B1 at Amud are the most varied; samples R\_1024 and R\_1026 have highly elevated 87Sr/86Sr ratios compared to samples
 R\_1025 and R\_1027. Unit B at Amud is composed of interbedded calcareous silt, calcareous concretions and ash derived from anthropogenic activity (Madella et al., 2002). These elevated 87Sr/86Sr ratios are likely due to the ash component in these sediments, and the variable nature of the interbedded sediments at this site.

**195 4.3 Comparison with regional datasets**

Several previous studies of bioavailableanalysing strontium isotopes have been undertaken in Israel, particularly in Northern Israel and the Golan Heights region. The results obtained in this study of soils derived from basalts in the Golan Heights region have 87Sr/86Sr valuesratios in the range of 0.70577270577 to 0.70681070681 for samples (sample numbers: 175, 203, 204, 205 and 216) with robust geological provenance. Basalt rock samples (sample numbers: 119, 143, 144, 147 and 278) in this study from the Golan Heights region have 87Sr/86Sr valuesratios ranging from 0.70529170529 to 0.70680770681. For the same region, Shewan (2004) report 87Sr/86Sr valuesratios ranging from 0.70529 to 0.70571 (n=4), while the results of Spiro et al. (2011) of 87Sr/86Sr measured from water samples in the Hula Valley and Golan Heights region range from 0.70467 to 0.70790 (n=37). The small disparitydifference between the results of this study and those of Shewan (2004) may reflect the limited number of samples analysed, as the larger data set of Spiro et al. (2011) encompasses the results of both this study and Shewan

- 205 (2004). Hartman and Richards (2014Weinstein (2006) measured bedrock 87Sr/86Sr from basalt units in the region, ranging from 0.7031 to 0.70337034, lower than those measured in this study. Hartman and Richards (2014) also-measured 87Sr/86Sr valuesratios from plants and invertebrates, and for samples taken from areas with basalt bedrock, ligneous (woody) plants were found to have 87Sr/86Sr valuesratios ranging from 0.70455870456 to 0.70851370851, non-ligneous plants ranging from 0.70472870473 to 0.70872, and invertebrates from 0.70494 to 0.70867770868. Rosenthal et al. (1989) summarise Sr87Sr/86Sr
- 210 isotope sampling of basalt derived groundwater in the Jordan Valley, with 87Sr/86Sr valuesratios of 0.7045 to 0.705. Rainwater in the Golan Heights region was found to have a Sr87Sr/86Sr isotope composition in the range of 0.70804 to 0.70923 (Herut et al., 1993). The wide range of Sr87Sr/86Sr isotope results in the Golan Heights region may be explained by the rapid depletion of Sr from rocks and soil in this region due to weathering and the large contribution of aeolian dust to soil profiles (Singer, 2007: 202-206).
- 215 Carbonate units were the most sampled lithology in this study, due to their widespread distribution across the country. This lithology groups a range of carbonate units (limestones, dolostones, chalk and marl), and the 87Sr/86Sr <del>valuesratios</del> of rock samples range from 0.70732870733 to 0.70911270911, while soil 87Sr/86Sr <del>valuesratios</del> range from 0.70802570803 to 0.71019971020. Hartman and Richards (2014) report bedrock 87Sr/86Sr <del>valuesratios</del> from Northern Israel for carbonate units to range from 0.7073 to 0.7078, comparable to the lower <del>values in the</del> range of rock samples measured in this study. Ligneous
- 220 plant samples have 87Sr/86Sr valuesratios ranging from 0.70788970789 to 0.70909570910, non-ligneous plants from 0.70791 to 0.70918170919 and invertebrates from 0.70806470807 to 0.70876170876 (Hartman and Richards, 2014), comparable to the

results of this study from rock and soil samples. Rosenthal et al. (1989) summarise  $S \pm^{87} Sr^{/86} Sr$  isotopes from carbonate derived water sources in the Jordan Valley as having 87Sr/86Sr values ratios ranging from 0.7070 to 0.7080.

- Arnold et al. (2016) sampled plants in the vicinity of Tell es-Safi/Gath to create a local bioavailable strontium map as a baseline 225 for use in interpreting the mobility of domestic animals from archaeological sites in the region. Of the 10 plant samples collected, four were from kurkar soils, with 87Sr/86Sr <del>values</del>ratios ranging from 0.708631</del>70863 to 0.70881070881. The kurkar soil 87Sr/86Sr valuesratios reported in this study are slightly more elevated, ranging from 0.70894470894 to 0.70934470934. Hartman and Richards (2014) also measured 87Sr/86Sr values ratios from plants and invertebrates from kurkar units, ligneous plants ranged from 0.70899270899 to 0.70902770903, non-ligneous plants ranged from 0.70899870900 to 0.70903470903,
- 230 and two mollusc shells were measured with 87Sr/86Sr valuesratios of 0.70896870897 and 0.70904370904, 
[revised manuscript text omitted]

- 325 KromMadella, M. D., Cliff, R. A., Eijsink, L., Jones, M., Herut, B.Goldberg, P., Goren, Y. and Chester, RHovers, E.: The characterisationExploitation of Saharan dusts and Nile particulate matterPlant Resources by Neanderthals in surface sedimentsAmud Cave (Israel): The Evidenve from the Levantine basin using Sr isotopes, MarPhytolith Studies, J. Archaeol. Sci., 29(7), 703-719. Geol., 155(3-4), 319-330, doi: 10.1016/S0025-3227(98)00130-3, 19991006/jasc.2001.0743, 2002.
- Maurer, A. F., Galer, S. J. G., Knipper, C., Beierlein, L., Nunn, E. V., Peters, D., Tütken, T., Alt, K. W. and Schöne, B. R.: Bioavailable87Sr/86Sr in different environmental samples - Effects of anthropogenic contamination and implications for isoscapes in past migration studies, Sci. Total Environ., 433, 216–229, doi:10.1016/j.scitotenv.2012.06.046, 2012.
   McDermott, F. and Hawkesworth, C.: The evolution of strontium isotopes in the upper continental crust, Nature, 344, 850-

853, 1990.

[revised manuscript text omitted]